# Microgravity Impacts the Expression of Aging-Associated Candidate Gene Targets in the p53 Regulatory Network

**DOI:** 10.3390/ijms262211140

**Published:** 2025-11-18

**Authors:** Nik V. Kuznetsov, Daria D. Vlasova, Anastasia A. Kotikova, Elena Tomilovskaya, Milos Ljubisavljevic

**Affiliations:** 1Karolinska Institutet, Nobels väg 16, SE-17177 Stockholm, Sweden; 2ASPIRE Precision Medicine Research Institute Abu Dhabi, United Arab Emirates University, Al Ain P.O. Box 15551, United Arab Emirates; milos@uaeu.ac.ae; 3Institute of Biomedical Problems, Russian Academy of Sciences (IBMP RAS), 123007 Moscow, Russia; dlokatosh@gmail.com (D.D.V.); finegold@yandex.ru (E.T.); 4Biochemistry and Molecular Biology, Pirogov Russian National Research Medical University, 117997 Moscow, Russia; 5Department of Physiology, College of Medicine and Health Sciences, United Arab Emirates University, Al Ain P.O. Box 15551, United Arab Emirates

**Keywords:** aging, microgravity, dry immersion, gene expression, gene targets, p53 network, transcriptome

## Abstract

The extreme space environment accelerates aging and compromises human health. NASA has named five main hazards in space, including gravity changes. However, the contribution of each factor to the overall impact on biomolecular and cellular processes is not always clear. We aimed to explore the effects of microgravity on the transcriptomes of healthy volunteers, with a focus on gene expression in p53 pathways. Ten healthy men were exposed to dry immersion simulated microgravity (DI-SMG) for three weeks and blood samples were collected at five timepoints before, during and after the course of DI-SMG. T cells were purified from the peripheral blood samples and total RNA was isolated and sequenced followed by a bioinformatics analysis of the volunteers’ global transcriptomes. A differential expression of p53 network genes was observed. The expression of 30 genes involved in the p53 gene network was affected during a 3-week course of DI-SMG including classic p53 downstream target genes involved in cellular senescence: *GADD45*, *p21*, *PUMA*, *IGF1* and other target genes. For the first time, the p53-associated cell signaling pathways and gene networks in human T cells were reported to be affected in vivo by DI-SMG. It is evident that the relatively mild effects of simulated weightlessness on the human body are sufficient to activate these pathways. Identified transcriptomic changes point toward a potential molecular overlap with aging and cellular senescence. These findings could contribute to a broader research landscape that may lead to the discovery of a new class of drugs—MG-senolytics.

## 1. Introduction

The physiological changes caused by spaceflight are similar to those that occur with aging [1,2]. NASA’s Human Research Program has identified five hazards that astronauts will face in space. These include space radiation, isolation and confinement, distance from Earth, gravity (and the lack of it), and closed or hostile environments [3]. The use of a DI-SMG system is a well-established, ground-based approach to the modelling of orbital space microgravity in vivo. It provides investigators with unique data on the physiological effects induced by DI-SMG [4]. The knowledge gained from those investigations is important for developing countermeasures against undesirable effects caused by the space environment and aging [1]. The well-studied physiological impacts of microgravity on human organ systems still lack precise and thorough complements at the level of cellular effects and biomolecular response mechanisms to weightlessness.

The most commonly mutated gene in cancer [5], the *TP53* tumor suppressor gene, which encodes transcription factor p53, has been called the “guardian of the genome” by one of its discoverers, Professor Sir David Lane [6], for its cell protective function under conditions of DNA damage stress. The p53 gene network compiles multifunctional biomolecular players, including target genes downstream of p53 [7,8,9,10,11,12,13,14]. The p53 gene network responds to a variety of environmental and internal signals and comprises several crucial pathways including DNA repair, the regulation of cell cycle and the induction of cell cycle arrest, apoptosis and cellular senescence. These pathways are vital for maintaining genome integrity and cellular homeostasis. p53 and several members of the p53 gene network are considered to be involved in aging processes or mediating longevity [15,16,17].

A four-fold increase was reported in the content of p53 in the skin cells of rats in space on the STS-58 mission (Columbia) [18]. Furthermore, microgravity effects on the p53 pathway in vitro were registered in mouse sperm cells [19], cultured human lymphoblastoid cell lines [20,21], in macrophages [22] and in human soleus muscle after 3-day dry immersion [23]. In human lymphocytes from two healthy donors, exposure to the spaceflight environment aboard the ISS for 48 h resulted in an increase in *TP53* mRNA levels, an effect attributed to the combinatory action of space radiation and microgravity [24].

### Objective

In a DI-SMG study conducted on ten healthy volunteers, we aimed to investigate whether simulated microgravity affects the expression of aging-related genes in the p53 gene network in human peripheral blood T cells in vivo.

## 2. Results and Discussion

### 2.1. RNA-Seq Data Output Summary

The DI-SMG study was conducted and global transcriptome profiling via RNA-seq was performed on T cells as described below (see Section 3. Materials and Methods). RNA-seq analysis yielded expression data for 58,676 transcripts, including 17,760 known genes and 38,430 transcripts associated with uncharacterized gene products. This dataset has been deposited in the NCBI Gene Expression Omnibus (GEO) repository under accession number GSE301964. Subsequent analysis focused specifically on interrogating the p53 gene network.

### 2.2. TP53 Tumor Suppressor Gene Expression

The transcript level of the *TP53* tumor suppressor gene varied among all 10 volunteers in the study. At different timepoints of the experiment, *TP53* was up-regulated in 9 of 10 volunteers compared to background levels (timepoint “−7 day”, Figure 1), and the dynamics of its expression varied between individuals. Seven volunteers (## 1–5, 7, and 8) demonstrated an increase in *TP53* transcript levels at 7 days in SMG and then showed variable expression with an overall upward trend. Volunteer #6 showed a slight decrease in *TP53* transcript levels during the SMG course and then an up-regulation above the background (BG) level at the 28-day timepoint after the SMG course. Volunteer #9 showed an overall down-regulation of the *TP53* transcript level with a decline in its expression at 7 days and 14 days in SMG, and then its level was partially recovered at the 21-day timepoint. Volunteer #10 also showed a decrease in *TP53* expression at 7 days and 14 days in SMG, but it was increased on day 21 of SMG, and the expression level increased further after the SMG course on day 28 (Figure 2).

The statistical analysis reveals a statistically significant upward trend in *TP53* gene expression over the course of the study (Appendix A). This observation is supported by a mean slope coefficient of 1.22 ± 0.88 with a 95% confidence interval that does not include zero

### 2.3. p53 Gene Network

The expression profile of the p53 gene network was significantly altered in response to the 3-week DI-SMG intervention. The expression profile of the complete gene set (n = 75) from the KEGG p53 signaling pathway (hsa04115) was evaluated. Notably, a subset of 30 genes (Table 1) showed differential expression in all, or nearly all, of the ten volunteers. This gene set included central apoptosis and senescence regulators such as *GADD45*, *p21*, *PUMA*, *NOXA*, and *p19ARF*, highlighting the pathway’s central role in the physiological response to simulated microgravity.

In relation to the regulation of the cell cycle, the gene expression of three B cyclins (*CCNB1*, *CCNB2*, and *CCNB3*) and two cyclin-dependent kinase inhibitors (*CDKN1A* and *CDKN1B*) in p53 networks exhibited dynamic changes under DI-SMG conditions in all ten subjects.

Growth arrest and DNA damage-inducible alpha, beta and gamma (*GADD45A*, *GADD45B*, and *GADD45G*) playing roles in cell cycle arrest, DNA repair, apoptosis, innate immunity, genomic stability, and senescence showed coordinated differential expression during the DI SMG study in all ten volunteers (Figure 3).

KEGG pathway enrichment analysis indicated that DI-SMG exposure significantly affected the p53 signaling pathway hsa04115 (Appendix A).

### 2.4. Aging-Associated Genes in p53 Gene Network and Interacting Pathways

The differential expression of p53 network genes linked to aging and cellular senescence was recorded in the SMG study. The expression of dominant active p53 leads to the constitutive expression of downstream target genes and results in premature aging [15]. There were varying changes in the TP53 transcript level observed during the experiment as described above (Section 2.2, Figure 2).

The cell cycle cyclin-dependent kinase inhibitor 1A (CDKN1A) interacts with cyclin-dependent kinases CDK2 and CDK4 and blocks their progression through the cell cycle. BH domain Bcl-2-associated protein X (Bax) and p53 up-regulated modulator of apoptosis (PUMA) are potent inducers of apoptosis. Cellular senescence and apoptosis prevent tumorigenesis. Also, both pathways have the potential to deplete stem and progenitor cell pools that leads to impaired tissue renewal, ultimately impairing organ homeostasis—a hallmark of aging [17]. Genes encoding cyclin-dependent kinase inhibitors *CDKN1A* and *CDKN1B*, *Bax*, *PUMA*, and cyclin-dependent kinases *CDK4* and *CDK6* demonstrated differential expression during the DI SMG experiment in all ten volunteers. Sestrins are conserved stress-inducible anti-aging genes [25]. The expression of genes encoding sestrins was changed during DI SMG in all or almost all ten volunteers (Figure 3, Table 1).

In addition, p53 affects the insulin-like growth factor 1 (IGF1) signaling pathway, which is a key determinant of aging and longevity [17]. Reduced IGF1 signaling is associated with extended lifespan in the highly conserved pathway among different organisms from nematodes to mammals [26]. IGFBP-3 plays an important role in senescence as an aging marker. It binds to IGF1, which regulates growth, survival, and aging. IGFBP-3 up-regulates the PI3K/Akt/mTOR signaling pathway during cell aging [27]. Also, p53 was elevated in *IGFBP3* gene KO cells when compared to normal cells [28]. The expression of both genes: *IGF1* and *IGFBP3* was changed during the DI SMG study in 5 out of 10 and in all 10 volunteers, respectively (Figure 3, Table 1).

**Table 1 ijms-26-11140-t001:** p53 pathway genes differentially expressed in 10 volunteers’ T cells during 3-week DI-SMG study.

#	NCBI Gene ID	Gene Symbol	Timepoint/ log2FC/ q-Value	Gene Name	Cite
1	317	*APAF1*	14 day/0.9944/0.0115	apoptotic peptidase activating factor 1	[29]
2	581	*BAX*	7 day/0.6738/0.0464	BCL2-associated X, apoptosis regulator	[30]
3	27113	*BBC3*	7 day/1.0363/0.0245	BCL2 binding component 3	[31]
4	891	*CCNB1*	21 day/−1.1572/0.0027	cyclin B1	[32]
5	9133	*CCNB2*	21 day/−2.1904/0.0038	cyclin B2	[33]
6	85417	*CCNB3*	7 day/−2.3464/0.0014	cyclin B3	[34]
7	3732	*CD82*	14 day/0.6182/0.0440	CD82 molecule	[35]
8	983	*CDK1*	14 day/−1.6466/0.0474	cyclin-dependent kinase 1	[36]
9	1019	*CDK4*	7 day/1.1546/0.0060	cyclin-dependent kinase 4	[37]
10	1021	*CDK6*	14 day/−1.355/3.34 × 10^−6^	cyclin-dependent kinase 6	[38]
11	1026	*CDKN1A*	7 day/−5.2107/2.27 × 10^−32^	cyclin-dependent kinase inhibitor 1A	[39]
12	1026	*CDKN2A*	21 day/1.8168/0.0341	cyclin-dependent kinase inhibitor 2A	[40]
13	54205	*CYCS*	7 day/−1.4602/3.51 × 10^−8^	cytochrome c, somatic	[41]
14	1647	*GADD45A*	7 day/−3.9401/2.21 × 10^−37^	growth arrest and DNA damage-inducible alpha	[42]
15	4616	*GADD45B*	7 day/−2.3869/3.54 × 10^−32^	growth arrest and DNA damage-inducible beta	[43]
16	10912	*GADD45G*	28 day/−3.1241/4.80 × 10^−3^	growth arrest and DNA damage-inducible gamma	[44]
17	51512	*GTSE1*	28 day/−1.9802/1 × 10^−5^	G2 and S-phase expressed 1	[45]
18	3479	*IGF1*	28 day/1.0213/0.0041	insulin-like growth factor 1	[46,47]
19	3486	*IGFBP3*	7 day/0.8320/0.0199	insulin-like growth factor binding protein 3	[48]
20	4194	*MDM4*	7 day/−0.7708/0.0022	MDM4 regulator of p53	[49]
21	5366	*PMAIP1*	7 day/−2.4366/2.24 × 10^−11^	phorbol-12-myristate-13-acetate-induced protein 1	[50]
22	27244	*SESN1*	14 day/0.9173/0.0069	sestrin 1	[51]
23	2810	*SFN*	7 day/−2.5663/4.25 × 10^−5^	stratifin	[52]
24	6477	*SIAH1*	7 day/−2.134/0.0121	siah E3 ubiquitin protein ligase 1	[53]
25	55240	*STEAP3*	21 day/−3.4598/0.0452	STEAP3 metalloreductase	[54]
26	7057	*THBS1*	7 day/−5.8423/4.97 × 10^−13^	thrombospondin 1	[55]
27	7157	*TP53*	21 day/1.1392/0.00134	tumor protein p53	[56,57]
28	63970	*TP53AIP1*	28 day/0.89438/0.0434	tp53-regulated apoptosis-inducing protein 1	[58]
29	9540	*TP53I3*	14 day/1.2081/0.0103	tumor protein p53-inducible protein 3	[59]
30	7161	*TP73*	14 day/−2.3214/0.0387	tumor protein p73	[60]

A number of key genes and encoded proteins, previously shown to be linked to aging, longevity and cellular senescence, are associated with p53 network genes and corresponding proteins as indicated by STRING (Figure 4). Among them are important regulators of cell signaling pathways: proto-oncogenes *Myc* [61] and *Jun* [62]; members of the NF-kB network (*NFKB1*, *NFKB2*, and *NFKBIA*) [63,64]; hypoxia-inducible factor 1-alpha (*HIF1A*) [65]; matrix metalloproteinase-9 (*MMP9*) [66]; and forkhead box P1 (*FOXP1*) [67].

### 2.5. Limitations of the Study and Future Perspectives

This study is subject to several limitations. Firstly, the cohort consisted of a small sample size (n = 10) of young, healthy male volunteers. Consequently, the findings may lack generalizability to female, older, or clinically compromised populations. Secondly, while the three-week DI-SMG protocol effectively models short-duration spaceflight, it may not fully recapitulate the physiological adaptations associated with longer mission or cumulative exposures. Finally, although microgravity was the variable of interest, the potential confounding influences of stress, physical inactivity, and confinement cannot be entirely discounted. To isolate the specific effects of microgravity simulation, future investigations should incorporate a parallel control group, such as subjects undergoing horizontal bed rest without SMG intervention. The transcriptomic changes identified here suggest a potential association with senescence. The direct measurement of senescence hallmarks, such as the secretion of SASP factors and the assessment of telomere attrition, will be critical for confirming the functional onset of senescence in future studies.

## 3. Materials and Methods

### 3.1. Dry Immersion Study Ethical Approval

All participated volunteers provided informed consent for the use and sharing of their fully anonymized data, according to the Helsinki Code of Medical Ethics for human samples. The Biomedical Ethics Committee of the IBMP RAS and Section of Physiology of the Bioethics Committee of the UNESCO National Bioethics Commission approved this study (Meeting No. 483 took place on 3 August 2018).

### 3.2. Dry Immersion Experiment Setup

An experiment with three weeks of exposure to DI-SMG without any countermeasures was performed at the Institute of Biomedical Problems (Figure 1A) during a period of 8 months from September 2018 to April 2019 with the participation of ten healthy men aged from 24 to 32 years as described [68].

### 3.3. Peripheral Blood Sample Collection

In total, five samples of 10 mL peripheral blood were collected in sodium heparin tubes from each volunteer at five timepoints during the time course of the DI SMG study: at 7 days prior to Day 0 of DI (timepoint “−7 day”; background, BG), after 7 days, 14 days and 21 days in DI SMG, and on the 7th day after DI SMG (timepoint “28 days”) (Figure 1B).

### 3.4. CD3+ T Cell Isolation

Peripheral blood mononuclear cells (PBMCs) were obtained using the Ficoll density gradient centrifugation isolation protocol optimized for Ficoll-Paque™ PLUS (Cytiva, Uppsala, Sweden). Human T cells were enriched by the positive CD3 cell selection kit (EasySep™ Human CD3 Positive Selection Kit II, Stem Cell Technology, Vancouver, British Columbia, Canada) with a purity of about 95%.

### 3.5. RNA Extraction

Total RNA was purified from sorted cells using the RNeasy kit and treated with the RNAse-free DNAse kit (##74104; 79254 QIAGEN, Hilden, Germany). The total RNA concentration was measured (NanoDrop2000, Thermo Scientific, Wilmington, DE, USA) and RNA integrity and purity were evaluated by gel electrophoresis analysis (1 × TAE 1% UltraPureAgarose, #16500500 Invitrogen, Waltham, MA, USA). Gel images were created with the ImageQuant LAS 4000 LAS4000 Image system using ImageQuant LAS 4000 Control Software (GE Healthcare, Freiburg, Germany) as described [69]. All selected isolated RNA samples were subjected to and passed both the internal lab quality control (QC) and QC performed by a sequencing company (Novogene, Hongkong).

### 3.6. RNA-Seq and Bioinformatic Analysis

Sequencing libraries were constructed from ribodepleted RNA using a stranded protocol. RNA sequencing and data quality control were performed with the Illumina HiSeq-PE150 Platform using the HiSeq 2500 Sequencing System at Novogene (Hongkong). Sequences were mapped to the reference genome with Tophat, v.2.0.12. Reads were aligned to the human reference genome assembly in December 2013 (GRCh38/hg38). The quality of the resulting RNA-seq data was assessed, with key metrics summarized in Appendix A. All bioinformatic software applications, their versions and statistical parameters are listed in Appendix A. The method, including software and statistical parameters, for differential gene expression analysis is detailed in Appendix A. In gene expression analysis by Novogene (Hongkong), H-cluster, K-means, and SOM were used to cluster log2 (ratios). Genes with an adjusted *p*-value (q-value) of <0.05 and an absolute log2 fold change of >1 were considered significantly differentially expressed. The gene expression heatmap was composed in NASQAR v1.0 (New York University (NYU) Center for Health Informatics and Bioinformatics, New York, NY, USA) using log2 normalized count matrix RNA-seq reads. The functional classification of refined data for peak-related genes was performed using NCBI Gene resource www.ncbi.nlm.nih.gov (accessed on 22 June 2025) (Bethesda, MD, USA), as described previously [70,71,72].

### 3.7. Statistical Analysis

A statistical analysis was performed to highlight the p53 gene expression trend over time course of the DI-SMG study. A linear trendline was calculated for each subject, and the resulting slope coefficients were analyzed using Excel (Microsoft 365 version (Microsoft Corporation, Redmond, WA, USA). The statistical inference of a significant upward trend was based on the statistical property that the 95% confidence interval around the mean slope was greater than zero.

### 3.8. Protein–Protein Interaction (PPI) Network

A network of protein–protein interactions (PPIs) was constructed using STRING (v12.0, Swiss Institute of Bioinformatics (SIB), Lausanne, Switzerland; string-db.org accessed on 19 September 2023) with all active evidence channels enabled, including Experiments, Databases, Co-expression, Neighborhood, Gene Fusion, Co-occurrence, and Textmining, and interactions were filtered to include only those with a high confidence score of ≥0.700; subsequently, KEGG pathway enrichment analysis was performed, with statistical significance defined by a corrected *p*-value of <0.05 using the Bonferroni method. The PPI network was generated with the input gene set comprising molecular components annotated in the KEGG p53 signaling pathway (hsa04115). The combined confidence scores, which integrated evidence from multiple channels, were calculated by STRING using a default phylogenetically curated collection of genomes as the statistical background.

## 4. Conclusions

The principle of design, uniqueness and robustness of the dry immersion system allows for original research to be carried out in simulated microgravity conditions close to orbital ones. T cell response to simulated microgravity includes the differential expression of p53 network aging-associated genes as well as interconnections with key regulators in interacting pathways. This finding opens an interesting opportunity to explore new protective strategies, which could, in time, provide insights into the future development of senolytics.

## Figures and Tables

**Figure 1 ijms-26-11140-f001:**
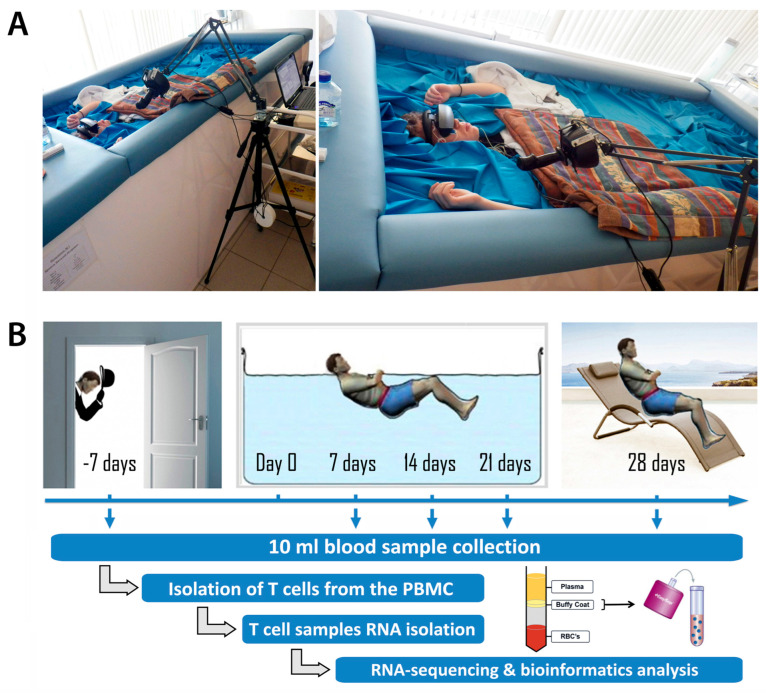
The setup of the simulated microgravity study in the dry immersion system. (**A**). The dry immersion facility at the Institute of Biomedical Problems (**B**). The time course and flow chart of the DI-SMG study. Time course timepoints: before (−7 day or BG, background), during (7 day; 14 day; 21 day) and after (28 day) 3 weeks of DI-SMG.

**Figure 2 ijms-26-11140-f002:**
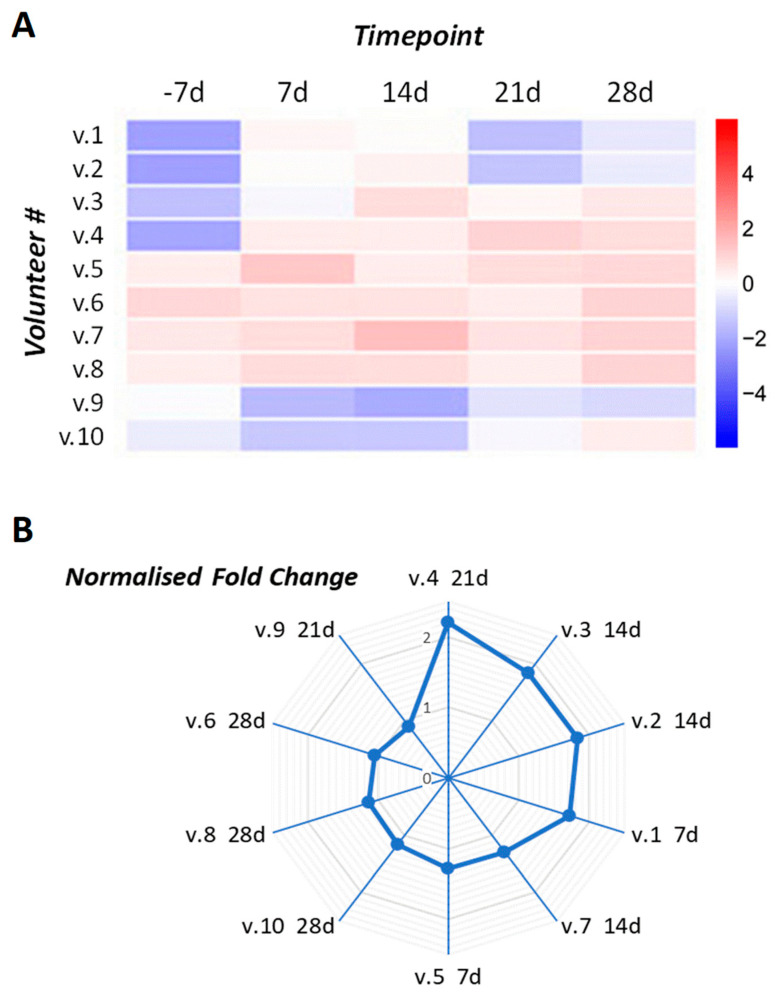
*TP53* gene differential expression in the DI-SMG study. A general trend towards an increased expression of the tumor suppressor gene *TP53* was observed in T cell samples from 9 of 10 volunteers. Time course timepoints: before (−7 day or BG, background), during (7 day; 14 day; 21 day) and after (28 day) 3 weeks of DI-SMG. The parameters shown are (**A**) non-normalized expression values, Log_2_ fold change; (**B**) normalized fold change. # is the number of volunteers.

**Figure 3 ijms-26-11140-f003:**
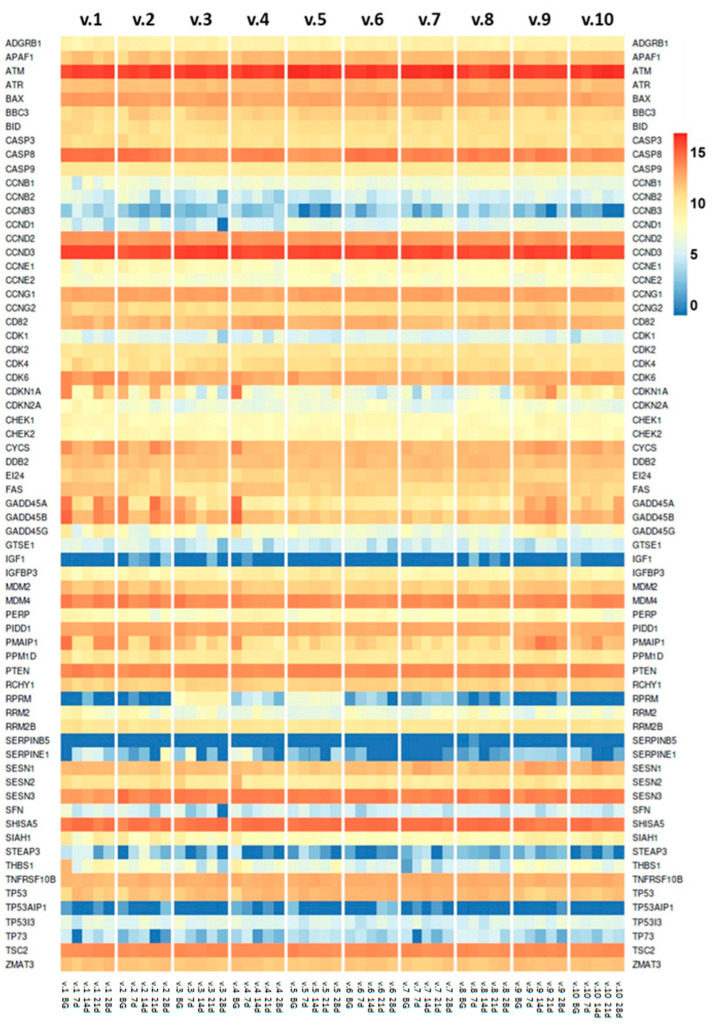
A heatmap of p53 gene network expression in the DI SMG study. The expression of the genes involved in the p53 gene network in ten volunteers’ T cells at five timepoints: before (BG, background or −7 day), during (7 day; 14 day; 21 day) and after (28 day) 3 weeks of DI-SMG. The log_2_ normalized count matrix RNA-seq read values are shown.

**Figure 4 ijms-26-11140-f004:**
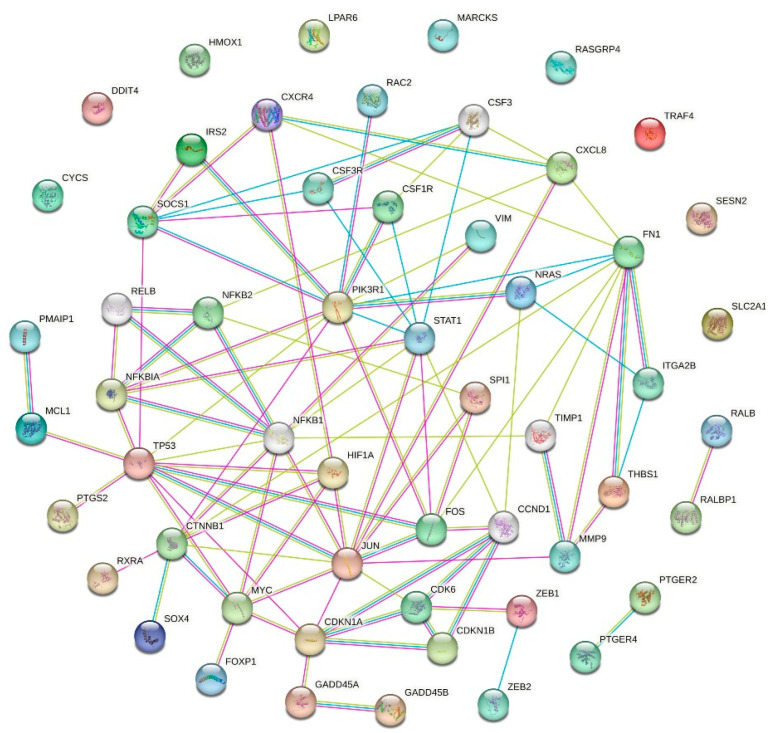
The protein–protein interaction (PPI) network of the p53 signaling pathway. The PPI network was generated using the STRING database, with the input gene set comprising molecular components annotated in the KEGG p53 signaling pathway (hsa04115). The network illustrates the complex functional relationships between the components of this key pathway, which was significantly enriched in the DI-SMG study, and key regulators in interconnecting signaling pathways. Known (blue and purple edges) and predicted (green edges) interactions are shown.

## Data Availability

The data that support the findings of this study are currently available in the NCBI GEO public repository for the editorial board and manuscript reviewers only. https://www.ncbi.nlm.nih.gov/geo/ [Gene Expression Omnibus] [https://www.ncbi.nlm.nih.gov/geo/query/acc.cgi?acc=GSE301964 (accessed on 22 June 2025)] [GSE301964].

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
