# Peer review of "Microgravity Impacts the Expression of Aging-Associated Candidate Gene Targets in the p53 Regulatory Network"

_ijms, 2025, doi:10.3390/ijms262211140_

Round 1

Reviewer 1 Report

Comments and Suggestions for Authors

The authors have analyzed bulk RNA-seq from CD3⁺ T cells collected from 10 healthy men undergoing 21 days of dry-immersion (DI) simulated microgravity, sampled at −7, 7, 14, 21, and 28 days. The manuscript reports TP53 up-regulation in 9/10 volunteers and differential expression of “nearly 30” p53-network genes (e.g., GADD45A/B/G, CDKN1A/p21, BBC3/PUMA, NOXA, IGF1, IGFBP3). The authors concluded that DI alone is sufficient to activate p53-associated pathways in human T cells and speculates about “microG-senolytics.”

The longitudinal sampling is valuable. However, the analysis and reporting are presently too descriptive and under-specified statistically, and the manuscript draws causal and translational inferences (e.g., “microG-senolytics”) that outpace the evidence. With substantial revision and additional analyses, this could be a concise, informative Brief Communication.

Some comments/objections:

  1. The definition and evidence for “nearly 30 genes” are vague.
    List the exact genes, directions, time-points, and FDR values that meet prespecified criteria. Table 1 currently summarizes functions from the literature rather than measured effects in this dataset. Consider moving Table 1 to the Supplemental information and replacing it with a concise results table (gene, log2FC, FDR at 7/14/21/28 days).
  2. Attribution to “microgravity alone” needs tempering or controls.
    DI is a powerful model, but it entails confinement, sensory changes, fluid shifts, stress, and circadian disturbance—the manuscript itself notes that isolating contributions of spaceflight hazards is not always clear. Either include a parallel control (e.g., head-down bed rest, water immersion, or hydrodynamic stress comparator) or re-frame claims to “DI-SMG” rather than “microgravity alone.”
  3. Briefly mention the limitations of the study/approaches used for analysis in the Abstract and Discussion.
  4. Over-interpretation toward aging/“microG-senolytics.”
    Finding DE in p53-network and aging-related genes is intriguing, but no senescence or functional readout is provided (no SASP markers, telomere metrics, or protein-level corroboration).
  5. STRING network parameters.
    Report STRING version, evidence type, confidence cutoff, and multiple-testing handling for enrichment. Otherwise, the network is largely illustrative.
  6. Data availability & reproducibility.
    There is no GEO/SRA accession or code repository. Please deposit raw FASTQ, processed counts, and metadata (subject ID, time, RNA metrics) and share an analysis script (alignment/quantification, DESeq2 steps, figure code). If data were generated in 2018–2019, confirm consent covers data sharing.
  7. Cohort limitations and generalizability.
    The cohort is male-only, 24–32 years; note this prominently and avoid general claims about “human T cells” without sex/age caveats. Consider including sex as a limitation and, if feasible, referencing plans for validation in females/older adults.
  8. Abstract precision.
    Replace qualitative phrases (“nearly 30 genes,” “noticeably changed”) with quantitative statements (counts with FDR<0.05 and median |log2FC| thresholds). Ensure the Abstract discloses n=10, male-only, and DI-SMG model.

Minor comments (style, formatting)

  • Reporting details for RNA-seq. Library prep (stranded vs unstranded), ribo-depletion vs polyA, read depth per sample, alignment and mapping rates, annotation version, and count tool should be stated (currently “standard bioinformatics at vendor”).
  • Figure labelling. In Fig. 2, the time-axis labels render oddly (“- 7d 7d 14d 21d 28d”). Please fix spacing and add subject IDs in a consistent order.
  • Terminology consistency. Use one of DI, SMG, microG consistently (and define once). The Introduction toggles between terms.
  • Ethics and consent. Ethics approvals and consent are stated; please add the approval ID numbers/dates if available.
  • Table 1 utility. Consider moving functional summaries to Supplement and replacing with dataset-specific details (direction, time-point, FDR).

Reviewer 2 Report

Comments and Suggestions for Authors

In this work, the author investigates the effect of simulated microgravity on the network of genes dependent on the p53 global regulator. Such a study, which concerns only microgravity conditions, is conducted for the first time. For the work, 10 volunteers were selected, from whom blood was taken one week before the experiment, on days 7, 14 and 21 under microgravity conditions and on day 28 - one week after the end of the experiment. T cells were isolated from peripheral blood. Then, total RNA was extracted and sequenced, and the obtained data were subjected to bioinformatics analysis. Differential expression of both p53 itself and the associated and dependent genes of its regulatory network was shown. Statistically significant data were obtained with a high degree of reliability that the expression of p53 and associated genes as a whole increases under microgravity conditions. Only one of the 10 volunteers showed a stable decrease. Many of the genes included in the p53 network are associated with premature aging. Therefore, although each person's body is individual, which can be clearly seen in the example of volunteer #9, one can indeed draw conclusions from the work that microgravity itself can contribute to premature aging of a person in space flight conditions. The experimental design is simple, the research methods are widely used, and the description of the work allows the study to be repeated. The amount of information corresponds to the correctly chosen style of presentation - a brief communication. The reviewer has no serious comments on the text, methods, or results obtained.

Minor point

Line 114 – There is no such section (4.1) in the paper. Please correct.

Reviewer 3 Report

Comments and Suggestions for Authors

The manuscript by Nik V. Kuznetsov describes data on simulated microgravity and its impact for the p53 pathway. Despite potential interest, the study lacks clarity, careful description of methods and statistical analysis.

First of all, the author should pay attention to statistical evaluation of the results. For example, the data on p53 (Fig. 1) lack such an analysis. A clear effect of p53 increase on day X, compared to control, would look much better.

Second, the author should carefully describe the obtained data, and expression of how many genes could be analyzed. Otherwise the analysis of just selected p53-dependent genes and conclusion, that the p53-dependent genes change their expression is methodologically incorrect.

In addition, the author should pay attention to text similarity to other sources. Currently, text mach is 79% which is extremely high.

Reviewer 4 Report

Comments and Suggestions for Authors

This brief communication investigates the effect of simulated microgravity (SMG) on aging-associated gene expression, focusing on the p53 regulatory network in human T cells. Ten healthy male volunteers (ages 24–32) underwent a three-week dry immersion (DI) protocol, a validated ground-based model of microgravity. Peripheral blood was collected at five time points before, during, and after SMG exposure. Purified CD3+ T cells were subjected to RNA sequencing, and differentially expressed genes (DEGs) in the p53 pathway were analyzed.

Key findings include:

  • Upregulation of TP53 in 9 of 10 volunteers.

  • Differential expression of ~30 p53-network genes, including CDKN1A (p21), GADD45, PUMA, NOXA, and IGF1.

  • Alterations in genes involved in senescence, apoptosis, DNA repair, and the IGF1/IGFBP3 signaling axis.

  • Suggestion that SMG-induced p53 pathway activation may accelerate cellular aging processes.

  • Proposal of “microG-senolytics” as a novel therapeutic category.

Limitations:

  • Sample Size: Only 10 participants, all young healthy men. This limits generalizability to women, older populations, or those with pre-existing conditions.

  • Duration: A three-week SMG exposure may not fully capture long-term or cumulative effects relevant to spaceflight missions.

  • Control Group: No parallel non-SMG immobilization control (e.g., bed rest without immersion) to disentangle microgravity effects from stress, inactivity, or confinement.

  • Causality vs. Association: While p53 pathway changes were observed, attributing them solely to SMG without ruling out other physiological stressors is premature.

  • Overextension of Conclusions: The idea of “microG-senolytics” is speculative. Drug development implications require far stronger mechanistic evidence, including functional assays of senescence or apoptosis.

  • Clinical Relevance: It remains unclear how these findings translate to health risks in astronauts or to broader aging biology on Earth.

Round 2

Reviewer 3 Report

Comments and Suggestions for Authors

The revised version of the manuscript "Microgravity Impacts Expression of Aging-associated Candidate Gene Targets in the p53 Regulatory Network" includes a reply to my previous comments, of which main consern was about the absence of clear statistical evaluation of the data. The Author has added a second illustration to Fig 2 (previously I've mentioned the figure on p53, thank you for right understanding of its number). However, no statistical evaluation was added. I also would like to pay attention to the absence of corresponding section in M&M chapter. The only words "statistical" in the manuscript corresponds to the description of STRING. The words "significantly" corresponds to the data of Table 1, which has q-values, but lacks description of how the data were evaluated.

Starting from the description of Fig. 2 (p53) the Author should have illustrated a clear statistical effect. Currently, the visualized picture just show, that in 9 of 10 volunteers the p53 expression was elevated at least once. However, it also shows, p53 was decreased in 5 of 10 volunteers at least once, that is not speaking of day -7. One should have showed column graphs for a sample of 10 volunteers and use statistics to support the suggestions.

The use of STRING (Fig. 4) is not motivated, and the figure does not indicate the idea well. Importantly, the criteria for the subset of visualized genes are not clearly described. In addition, no results of the mentioned "KEGG pathway enrichment analysis" (Methods) are clearly presented. What was used for the background for such an analysis?

Taking into account the raised questions about statistics, and the fact that the text was not corrected accordingly, an extensive correction is recommended.

PS. The Editors of course can check the output of the Ithenticate report too.

Round 3

Reviewer 3 Report

Comments and Suggestions for Authors

The Author of the manuscript "ijms-3825190" has made a substantial effort to improve the clarity of the text and figures. Still, it is not clear enough, and unanswered questions to methodology and thus to the presented data arise.
As before, the main question is about the p53 and data presented as Fig. 2 and now also as Fig. S5. As I understand from the text (although it should be clearly stated either in the figure caption or/and in the corresponding "Methods" section) the BG symbol in Fig. S5 corresponds to "background levels (timepoint “-7d”, Figure 1)" - line 85 of the text. This makes calculations presented in Fig. S5 compromised, because the assumed linear trend does not take into account the distance between BG and the first point being twice longer (-7 and +7 days) than the other ones (7-days differences).
Moreover, inclusion of the point 28 days is very questionable according to the scheme (Fig. 1), and calculations without it looks as the first choice, because subjects experience no simulated microgravity for 7 days already at day 28. Since it is widely reported that p53 half life is about 20 min, the last point (28d) should be excluded.
All figures should clearly explain the depicted data. The "Y" axes in Fig. S5 are not labelled, which together with different corresponding values used for Fig. 2A and 2B makes it all confusing.
I've also mentioned before my concerns about the data analysis. As I've written before, "Otherwise the analysis of just selected p53-dependent genes and conclusion, that the p53-dependent genes change their expression is methodologically incorrect". The Author included a brief description for the obtained RNA-seq dataset (subsection 2.1.), however this small inclusion does not solve the question under consideration. For example, similar details and a clear description of data analysis in STRING is missing. It's not clear, what background and what list of genes were used. From what I can see, the data analysis procedure may be methodologically incorrect, but it's full description is missing to check it properly, which of course should be resolved.

Thus, as I've written before, the study still lacks clarity and careful descriptions of the methodology, data analysis and results.
